# Pharmaceutical cost dynamics for the treatment of rifampicin-resistant tuberculosis in children and adolescents in South Africa, India, and the Philippines

Thomas Wilkinson[1]*, Anthony J. Garcia-Prats[2,3], Tina Sachs[3], Mandar Paradkar[4,5], Nishi Suryavanshi[4,5], Aarti Kinikar[6], Melchor V. Frias[7], Edina Sinanovic[1], Anneke C. Hesseling[3], James. A. Seddon[3,8], Megan Palmer[3]

1 Health Economics Unit, School of Public Health, University of Cape Town, Cape Town, South Africa, 2 Department of Pediatrics, University of Wisconsin-Madison School of Medicine and Public Health, Madison, Wisconsin, United States of America, 3 Faculty of Medicine and Health Sciences, Department of Paediatrics and Child Health, Desmond Tutu TB Centre, Stellenbosch University, Stellenbosch, South Africa, 4 Byramjee Jeejeebhoy Government Medical College-Johns Hopkins University Clinical Research Site, Pune, Maharashtra, India, 5 Center for Infectious Diseases in India, Johns Hopkins India, Pune, India, 6 Department of Paediatrics Byramjee Jeejeebhoy Government Medical College and Sassoon General Hospitals, Pune, India, 7 De La Salle Medical and Health Sciences Institute Research Division, Dasmarinas Cavite, Philippines, 8 Department of Infectious Disease, Imperial College London, London, United Kingdom

* tommy.d.wilkinson@gmail.com

**Data Availability Statement:** All relevant data are within the manuscript and its Supporting Information files. There is an additional data tool

## Abstract

Rifampicin-resistant (RR) tuberculosis (TB) in children is a major global health concern but is often neglected in economics research. Accurate cost estimations across the spectrum of paediatric RR-TB treatment regimens are critical inputs for prioritisation and budgeting decisions, and an existing knowledge gap at local and international levels. This normative cost analysis was nested in a Phase I/II pharmacokinetics, safety, tolerability, and acceptability trial of TB medications in children in South Africa, the Philippines and India. It assessed the pharmaceutical costs of 36 childhood RR-TB regimens using combinations from 16 different medicines in 34 oral formulations (adult and child-friendly) in 11 weight bands in children <15 years of age. The analysis used local and Global Drug Facility pricing, and local and international guideline recommendations, including adaptions of BPaL and BPaLM regimens in adults. Costs varied significantly between regimen length, age/weight banding, severity of disease, presence of fluroquinolone resistance, and different country guideline recommendations. WHO recommended regimen costs ranged 12-fold: from US$232 per course (short regimen in non-severe disease) to US$2,761 (long regimen in severe, fluroquinolone-resistant disease). Regimen treating fluoroquinolone-resistant infection cost US$1,090 more than comparable WHO-recommended regimen. Providing child-friendly medicine formulations in <5-year-olds across all WHO-recommended regimens is expected to cost an additional $380 (range $212-$563) per child but is expected to have wider benefits including palatability, acceptability, adherence, tolerability, and dose accuracy. There were substantial differences in regimen affordability between countries when adjusted for purchasing power and domestic spending on health. Appropriate, effective, and affordable

that allows users to interact with the data, https://blogs.sun.ac.za/dttc/links/. The data tool does not provide any additional data that is not provided in the paper and supporting files.

**Funding:** TW, AJGP, TS, MP, NS, AK, MVF, AH, MP are part of the study team of the CATALYST trial, under the BENEFIT Kids project https://unitaid.org/project/better-treatments-and-prevention-for-drug-resistant-tuberculosis-in-children/#en. This project is made possible thanks to Unitaid's funding and support. Unitaid accelerates access to innovative health products and lays the foundations for their scale-up by countries and partners. The funder played no role in the study design, data collection or analysis, decision to publish, or preparation of the manuscript.

**Competing interests:** The authors have declared that no competing interests exist.

treatment options are an important component of the fight against childhood RR-TB. A comprehensive understanding of the cost and affordability dynamics of treatment options will enable national TB programs and global collaborations to make the best use of limited healthcare resources for the care of children with RR-TB.

## Introduction

There is increasing recognition of the importance of rifampicin-resistant (RR) tuberculosis (TB), defined as *Mycobacterium tuberculosis (M. tuberculosis)* resistant to at least rifampicin. An estimated 410,000 incident cases of RR-TB occurred in 2022 [1], and recent model-based estimates suggest that 25,000 to 32,000 new cases of RR-TB occur in children 0–14 years of age each year [2–4].

In contrast to adults with RR-TB, outcomes among children treated for RR-TB are generally good, with four out of five patients successfully treated [5]. Despite these good outcomes, RR-TB treatment for children remains lengthy (traditionally 9–18 months), complex, and has frequent and significant adverse effects. There have been substantial changes to treatment recommendations for RR-TB treatment in recent years, with the 2022 update of the WHO RR-TB treatment guideline recommending all-oral regimens for children in most cases, with injectable agents limited to salvage therapy only [6].

Treating RR-TB disease in children with shorter, safer, and more effective regimens will require the availability of existing and repurposed medicines, and newer agents such as bedaquiline and delamanid. In addition, the practicality of accurately and consistently administering medicines is improved with child-friendly formulations.

CATALYST was a multi-centre trial investigating the pharmacokinetics, safety, tolerability, and acceptability of new child-friendly formulations of moxifloxacin and clofazimine in children and adolescents routinely treated for RR-TB in South Africa, India, and the Philippines [7]. The CATALYST trial is completed with primary findings forthcoming. A secondary objective of the CATALYST trial was to describe the pharmaceutical costs of RR-TB treatment in children and adolescents. It is expected that any improvements in medicine formulations for children with RR-TB would be incorporated within clinical practice over time. However, unless the nature and profile of the costs of RR-TB are investigated more broadly, the financial implications of the adoption of new formulations for treatment of paediatric RR-TB remain uncertain.

An important initiative relating to RR-TB medicine availability and pricing is the Stop TB Partnership's Global Drug Facility (GDF), which aids global access to quality-assured TB medicines through a bundled procurement and supply mechanism [8]. Economies of scale mean that purchasing through the GDF may result in lower pricing and better supply chain certainty compared to individual country procurement mechanisms. The GDF catalogue includes formulations of TB medicines for adults and children, with pricing and formulation information regularly updated [9]. A United Nations High-Level Meeting on TB in 2018 encouraged all nations to use the GDF [10], and 134 countries accessed the GDF in 2023. Countries choose which medicines and formulations to procure through the GDF, for example a country may purchase routine first line TB medicines through national procurement mechanisms and utilize the GDF for supply of second line medicines with or without support of donors and development partners [11].

The cost to countries of different RR-TB treatment regimens in children can be uncertain due to the rapidly changing treatment landscape, variation in volumes for weight-based

dosing, and medicine price fluctuation (particularly for countries not using the GDF). This uncertainty may result in slow adoption of innovative approaches to care, especially in countries where the burden of RR-TB may be higher and affordability concerns more pronounced. There is currently limited empirical evidence on the cost of paediatric RR-TB care and how changes within the treatment regimen can affect constrained local and national budgets. Understanding the local cost dynamics associated with RR-TB medications and treatment regimens is essential to informing these procurement and implementation decisions.

The aims of this costing analysis were to (1) estimate the costs of a representative sample of RR-TB treatment regimens used in children and adolescents as recommended by global and national guidelines, (2) assess the in-country cost of the WHO-recommended RR-TB regimens in three high burden countries (South Africa, India, and The Philippines), and (3) explore the costs of country-specific RR-TB treatment regimens and local affordability.

## Methods

### Study design

We performed a pharmaceutical cost analysis, calculating the costs of normative treatment recommendations for children aged <15 years old across eleven weight bands (36 regimens, 16 medicines, 34 formulations) from the perspective of national TB programs in South Africa, India, and the Philippines.

The RR-TB treatment regimens included in this analysis reflected global (WHO [6], The Sentinel Project [12]) and national (South Africa [13], India [14], The Philippines [15]) guidance, and additional regimens based on author judgement. Regimens were grouped into six categories: 1) Short-duration standardised bedaquiline-containing regimens; 2) Long-duration fluroquinolone-susceptible (non-severe disease); 3) Long-duration fluroquinolone-susceptible, (severe disease); 4) Long-duration fluroquinolone-resistant, (non-severe disease); 5) Long-duration fluroquinolone-resistant, (severe disease); and 6) Adaptions of BPaL (bedaquiline, pretomanid, linezolid) and BPaLM (bedaquiline, pretomanid, linezolid, moxifloxacin) including substitution of pretomanid for delamanid. The regimens representing adaptions of BPaL/BPaLM are provided for comparison as potential future treatment scenarios, in line with experienced child TB physicians' opinion on potential treatment regimens available for children in the future [16,17].

Only all-oral treatment options were presented in this analysis as injectable-free regimens are the preferred approach for children in all settings reviewed, and it is now possible to build all-oral treatment regimens for all children with RR-TB due to the expanded use of bedaquiline and delamanid to children of all ages [1]. As the composition of RR-TB treatment regimens is largely the same for children living with or without human immunodeficiency virus (HIV), no separate analysis was conducted for children living with HIV. Further detail relating to the source recommendations are in Appendix 8 in S1 File.

### Data sources

The number of tablets and the approach to dosing of different formulations were extracted from WHO recommendations [18,19]. If dosing guidance differed between ages in the same weight band the mean of all the likely number of doses for that weight band was used.

TB medicine unit prices were sourced from the January 2024 GDF Medicines Catalogue [9]. Where the GDF catalogue reported a price range, the mid-point cost for that medicine was used. For the South African analysis, the February 2024 Master Health Product List (MHPL) [20] was used to extract TB medicine unit prices in South African Rand and converted to

**Table 1. Tuberculosis epidemiological profile: South Africa, India, the Philippines.**

| | South Africa | India | Philippines |
|---|---|---|---|
| Population (million) (2022)[1] | 59.4 | 1417.2 | 113.9 |
| Incidence of tuberculosis (per 100,000) (2022)[2] | 468 (304–665) | 199 (169–231) | 638 (337–1060) |
| Incidence of RR-TB: all ages, new cases (% of total incidence) (2022) [2] | 2.9% (2.8–3.0) | 2.5% (2.3–2.7) | 2.5% (2.0–2.9) |
| Proportion of TB case notifications aged 0–14 (%), (2022) [2] | 7% | 5% | 7% |
| TB treatment coverage (notified estimated incidence, 2022) [2] | 77% | 80% | 59% |
| TB treatment success rate, new and relapse cases (%) (2021) [2] | 79% | 87% | 80% |
| Household contacts of bacteriologically confirmed TB cases on TPT (%) (2022) [2] | 6% (5.8–6.1) | 22% (22–23) | 5.5% (5.-5.7) |
| Treatment success rate for RR/MDR-TB cases started on 2nd-line treatment (2020) [2] | 62% | 69% | 79% |

Source: 1. World Bank. World Development Indicators [22]; 2. Global Tuberculosis Report 2023 Tuberculosis country profiles [23–25].

TB: Tuberculosis; RR/MDR-TB: Rifampicin-Resistant or Multi-Drug Resistant Tuberculosis; TPT: Tuberculosis Preventive Therapy.

United States dollars (US$). When more than one supplier of medicine for the same formulation was available, the lowest priced formulation was used.

The Central TB Division in India conducts a competitive tender process; however, the specific pricing structures are not publicly available and were not made available following author requests. Given the high medicine volume purchased by the Central TB Division and extensive domestic manufacture of pharmaceuticals in India, GDF pricing was assumed to be indicative of prices in the Indian context for this analysis. GDF prices were used for analyses specific to the Philippines as the country accesses TB medications through the GDF. A medicine was considered child-friendly based on the GDF categorisation [21].

PPP conversion rates [21], exchange rates, and health financing indicators [22] were sourced from the World Bank DataBank. For country context, TB epidemiological profiles were sourced from the Global TB report 2023 Country TB profiles [23–25] (Table 1).

### Data analysis

Ages were matched to weight bands using WHO weight-for-age tables [26,27] to allow for the estimation of the proportion of children falling in the respective weight bands and assuming equal distribution between genders (Appendix 9 in S1 File for weight distributions across age ranges). This was required to calculate weight-based dosing costs for children in the <15 years and <5 years age groups, and to incorporate the uneven distribution of child weight (and therefore treatment costs) across age ranges. As children with RR-TB are commonly underweight-for-age, a -1 standard deviation was assumed to reflect weight-for-age based on Global Health Observatory population tables [28].

The costs of TB treatment regimens are represented in US$ and were calculated by multiplying medicine unit prices with the number of dosage units required for a regimen assuming a 4-week (28 day) treatment month.

WHO guidance was used to determine assumptions relating to wastage and the number of tablets or capsules that are required to achieve recommended doses [6]. Dispersible tablets and scored tablets can be halved, whereas unscored tablets and capsules cannot be halved, so any part of a tablet/capsule used to make up a child's dose represented a whole tablet/capsule used. The number of para-aminosalicylic acid sachets needed for the total daily dose was used to calculate the quantity needed per month (irrespective of whether daily doses were divided into two daily doses). It was assumed that when tablets or capsules are dissolved in water (or other dosing medium), the required dose is administered, and the remaining mixture discarded.

**Table 2. Health financing indicators related to TB in South Africa, India, and the Philippines.**

| | South Africa | India | Philippines |
|---|---|---|---|
| GDP per capita (US$ current) (2022)[1] | $6,776 | $2,411 | $3,499 |
| GDP per capita (PPP adjusted %I current) (2022) [1] | $15,920 | $8,400 | $10,137 |
| Current health expenditure (% of GDP) (2020) [1] | 9% | 3% | 5% |
| Domestic general government health expenditure, per capita (US$) (2020) [1] | $304 | $21 | $74 |
| Domestic general government health expenditure, per capita (PPP I$) (2020) [1] | $718 | $70 | $187 |
| Value of TB program/TB funding per incidence of TB (US$) [2,3] | $514 | $107 | $172 |
| Value of TB program/TB funding per incidence of TB ($I) [2,3] | $1,207 | $366 | $499 |
| Proportion of TB funding supported by external donors [2] | 17% | 19% | 15% |

Source: 1. World Bank. World Development Indi[cators [22]; 2. Global Tuberculosis Report 2023 Tuberculosis country profiles [23–25]; 3. Author calculation. GDP-gross domestic product, $I-international dollar, PPP-purchasing power parity, TB-tuberculosis, US$-United States Dollar.

The list of medicines and formulations included in the analysis is in S1 Appendix 1 in S1 File, along with the relevant wastage assumptions.

RR-TB treatment regimen costs were adjusted for purchasing power parity (PPP) in India, South Africa, and the Philippines. RR-TB regimen costs were analysed as a proportion of local TB budgets and total public health expenditure per person to gain insights into local affordability and to provide meaningful comparisons between countries. TB-related health financing indicators for India, South Africa and the Philippines are presented in Table 2.

## Results

The 36 regimens were clustered by regimen type, formulation used and age band. Child-friendly formulations were analysed for WHO and Sentinel Project regimens and adapted BPaL/BPaLM regimens only, given limited inclusion of child friendly formulations in national formularies and essential medicines lists South Africa, India, and the Philippines. Table 3 and Fig 1 details the total cost for each regimen and Appendix 3 in S1 File details the mean monthly costs for each regimen. An online interactive costing tool enables further representation of this analysis with user-specified changes to medicine and formulation choice and regimen length [29].

Overall, there is substantial variation in the costs of regimens based on age range, formulation type, treatment duration and regimen composition, driven by disease severity and presence of fluoroquinolone resistance.

### Standardised short all-oral regimens

**Bedaquiline-containing regimens.** The standardised "short" regimens (9–11 months) are currently the preferred choice of treatment for RR-TB according to WHO as well as South African and Indian national TB guidelines for children with uncomplicated, non-severe, fluoroquinolone-susceptible RR-TB disease. The regimens also had the lowest cost of those evaluated, across age bands and formulation type. The treatment duration is determined by culture conversion or health improvement at four months after treatment initiation, and two variations of the standardised course are recommended: one that includes linezolid (recommended in WHO and South African national guidelines) and another that includes ethionamide (recommended in WHO and Indian national guidelines).

Under GDF medicine pricing, the median cost of providing the short bedaquiline-containing RR-TB regimens (including either linezolid or ethionamide) was $157 (range $145 to

Table 3. Total regimen cost (US$) for RR-TB paediatric regimens (GDF prices).

| Regimen | Setting | Treatment duration (months) | Total regimen cost (US$) by age group and formulation | | | |
|---|---|---|---|---|---|---|
| | | | Adult formulations | | Child-friendly formulations | |
| | | | <15 years | <5 years | <15 years | <5 years |
| **SHORT BEDAQUILINE-CONTAINING ALL ORAL REGIMENS** | | | | | | |
| BDQ (6)-LFX-CFZ-Z-E-hH-ETO (4)/ LFX-CFZ-Z-E (5) | Global (WHO) India | 9 | 243 | 151 | 803 | 394 |
| LZD (2)-BDQ (6)-LFX-CFZ-Z-E-hH (4)/ LFX-CFZ-Z-E (5) | Global (WHO) South Africa | 9 | 232 | 145 | 784 | 382 |
| BDQ-LFX-CFZ-Z-E-hH-ETO (6)/ LFX-CFZ-Z-E (5) | WHO India | 11 | 286 | 175 | 945 | 466 |
| LZD (2)-BDQ-LFX-CFZ-Z-E-hH (6)/ LFX-CFZ-Z-E (5) | WHO South Africa | 11 | 265 | 164 | 902 | 441 |
| **LONGER ALL ORAL REGIMENS** | | | | | | |
| **Fluoroquinolone-susceptible–non severe** | | | | | | |
| LZD (2)-BDQ-LFX-CFZ (15) | Global (WHO) | 15 | 380 | 249 | 949 | 461 |
| BDQ-LFX-CFZ-CS (9) | Global (Sentinel Project) | 9 | 341 | 218 | 925 | 475 |
| LZD (2)-BDQ (6)-LFX-CFZ-TRD (15) | South Africa | 15 | 1,562 | 965 | - | - |
| LZD (2)-LFX-CFZ-TRD-PAS (15) | South Africa | 15 | 2,418 | 1,498 | - | - |
| BDQ-LZD (6)-LFX-CFZ-CS (15) | India | 15 | 484 | 302 | - | - |
| LZD-Z (6)-LFX-CFZ-CS (15) | India | 15 | 412 | 247 | - | - |
| LZD (6)-BDQ-LFX-CFZ-CS (12) | Philippines | 12 | 469 | 305 | - | - |
| LZD (6)-LFX-CFZ-CS-PAS (12) | Philippines | 12 | 1,077 | 672 | - | - |
| **Fluoroquinolone-susceptible–severe** | | | | | | |
| LZD (2)-BDQ-LFX-CFZ-CS (20) | Global (WHO) | 20 | 743 | 477 | 2,019 | 1,040 |
| LZD (2)-BDQ-LFX-CFZ-CS (12) | Global (Sentinel Project) | 12 | 455 | 293 | 1,244 | 640 |
| LZD (2)-BDQ-LFX-CFZ-TRD (18) | South Africa | 18 | 1,980 | 1,239 | - | - |
| LZD (2)-LFX-CFZ-TRD-PAS (18) | South Africa | 18 | 2,900 | 1,796 | - | - |
| LZD (6)-BDQ-LFX-CFZ-CS (18) | India | 18 | 684 | 442 | - | - |
| LZD (6)-LFX-CFZ-CS-Z (18) | India | 18 | 501 | 299 | - | - |
| LZD (6)-BDQ-LFX-CFZ-CS (18) | Philippines | 18 | 685 | 443 | - | - |
| LZD (6)-LFX-CFZ-CS-PAS (18) | Philippines | 18 | 1,605 | 1,000 | - | - |
| **Fluoroquinolone-resistant–non severe** | | | | | | |
| LZD (2)-BDQ-CFZ-CS (15) | Global (WHO) | 15 | 541 | 352 | 1,313 | 697 |
| BDQ-CFZ-CS-DLM (9) | Global (Sentinel Project) | 9 | 1,249 | 1,074 | 2,193 | 1,310 |
| LZD (2)-DLM (6)-BDQ-CFZ-TRD (15) | South Africa | 15 | 2,246 | 1,599 | - | - |
| LZD (2)-CFZ-TRD-DLM-PAS (15) | South Africa | 15 | 3,931 | 2,924 | - | - |
| LZD (6)-BDQ-CFZ-CS-DLM (12) | Philippines | 12 | 1,681 | 1,445 | - | - |
| LZD (6)-CFZ-CS-DLM-PAS (12) | Philippines | 12 | 2,288 | 1,813 | - | - |
| **Fluoroquinolone-resistant–severe** | | | | | | |
| LZD (2)-BDQ-CFZ-CS-DLM (20) | Global (WHO) | 20 | 2,761 | 2,378 | 4,837 | 2,895 |
| LZD (2)-BDQ-CFZ-CS-DLM (12) | Global (Sentinel Project) | 12 | 1,666 | 1,434 | 2,936 | 1,754 |
| LZD (2)-BDQ-CFZ-TRD-DLM (18) | South Africa | 18 | 3,796 | 2,950 | - | - |
| LZD (2)-CFZ-TRD-DLM-PAS (18) | South Africa | 18 | 4,716 | 3,507 | - | - |
| LZD (6)-BDQ-CFZ-CS-DLM (18) | Philippines | 18 | 2,502 | 2,154 | - | - |
| LZD (6)-CFZ-CS-DLM-PAS (18) | Philippines | 18 | 3,422 | 2,711 | - | - |
| **Adapted BPaL/BPaLM regimen** | | | | | | |

(*Continued*)

**Table 3.** (Continued)

| Regimen | Setting | Treatment duration (months) | Total regimen cost (US$) by age group and formulation | | | |
|---------|---------|------------------------------|--------|--------|--------|--------|
| | | | Adult formulations | | Child-friendly formulations | |
| LZD (2)—BDQ-DLM-LFX (6) | Global (potential future recommendation) | 6 | 783 | 699 | 1,496 | 854 |
| LZD (2)–BDQ-DLM-CFZ (6) | | 6 | 847 | 737 | 1,489 | 865 |
| BDQ-PA-LZD (6) | | 6 | 392 | 351 | 751 | 473 |
| BDQ-PA-LZD-MFX (6) | | 6 | 418 | 377 | 851 | 534 |

Costs reflect mean costs per regimen adjusted for distribution of weight-for-age in each category.

BDQ-bedaquiline, CFZ-clofazimine, CS-cycloserine, DLM-delamanid, E-ethambutol, ETO-ethionamide, hH-high dose isoniazid, LFX-levofloxacin, LZD-linezolid, MFX-moxifloxacin, PA-pretomanid, PAS-para-aminosalicylic acid, PTO-prothionamide, Pyr(B6)-Pyridoxine, TRD-terizidone, Z-pyrazinamide.

GDF-Global Drug Facility, US$-United States Dollar; WHO–World Health Organization; RR-TB–Rifampicin-resistant tuberculosis.

$175) in the <5-year-old age group when using adult formulations, and $417 ($382 to $466) when using child-friendly formulations.

**Longer all-oral regimens.** In most of the settings analysed, longer regimens were recommended for children not eligible to receive standardised WHO regimens. These regimens were broadly classified based on severity (typically requiring longer treatment periods) and fluoroquinolone susceptibility (limiting use of levofloxacin and moxifloxacin).

## Fluoroquinolone-susceptible, non-severe disease

Eight regimens were analysed with treatment durations ranging from 9 to 15 months. For children aged <15 years using adult formulations, monthly costs ranged from $25 to $161 ($341 to $2,418 for the total treatment period) for the WHO and South African-recommended regimens respectively. When using adult formulations, regimens containing terizidone ($84.69 per month) and/or para-aminosalicylic acid ($62.26 per month) were the most expensive, resulting in higher regimen costs in South Africa compared to recommendations from India, the Philippines or the WHO.

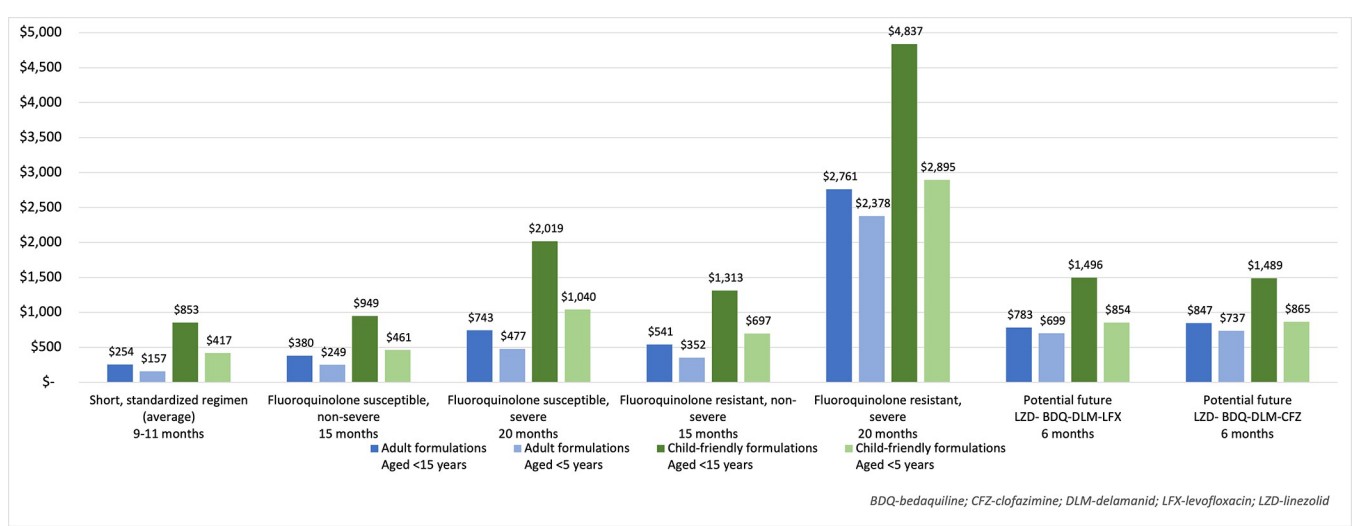

**Fig 1. Total regimen cost (US$): Who recommended RR-TB regimens by formulation and age range (GDF pricing).**

### Fluoroquinolone-susceptible, severe disease

A further eight regimens recommended for treatment of fluoroquinolone-susceptible, severe disease were analysed with treatment durations ranging from 12 to 20 months. The median treatment duration for severe disease was 3–4 months longer and cost 65.9% more than for non-severe disease across all regimens analysed. In common with non-severe disease, regimens containing terizidone and/or para-aminosalicylic acid were associated with the highest cost.

### Fluoroquinolone-resistant, non-severe disease

Six regimens were analysed with treatment durations ranging from 9 to 15 months for non-severe disease with fluoroquinolone resistance. Delamanid was the most expensive medicine in all but one of the regimens, contributing between 39% to 74% of overall regimen costs. The nine-month regimen recommended in the Sentinel Project is shorter than other guideline recommendations but had a higher medicine cost than the 15-month WHO-recommended regimen due to the inclusion of delamanid.

### Fluoroquinolone-resistant, severe disease

Six regimens were analysed with treatment durations ranging from 12 to 20 months. This category consistently incurred the highest costs due to treatment duration and inclusion of high-cost medicines. The recommended regimen within South African guidelines for this group includes delamanid, terizidone, and para-aminosalicylic acid given over a period of 18 months, and is expected to cost $4,716, with each month extension incurring an additional $262. This represents a 20-fold difference in costs between the least expensive regimen (9-month bedaquiline-containing all-oral regimen in uncomplicated cases) and the most expensive regimen for fluoroquinolone-resistant severe disease). WHO-recommended regimens for fluoroquinolone-resistant disease using adult formulations had a median cost approximately three times more than regimens used to treat fluoroquinolone-susceptible strains in severe and non-severe disease, equating to additional costs of $1,090 across the <15-year-old age group. This is mainly due to the exclusion of the relatively low-cost fluroquinolones (levofloxacin or moxifloxacin), requiring inclusion of higher cost delamanid and/or cycloserine. Inclusion of terizidone, para-aminosalicylic acid and bedaquiline also contributed substantially to the overall regimen costs.

**Adapted BPaL/BPaLM regimens.** Four regimens, reflecting the recent six-month BPaL and BPaLM regimen developments in adults, were included in the analysis for comparison, as future potential regimens to be used as further safety and efficacy data become available. The costs of the regimens using adult formulations in the <15-year-old age group ranged from $392 to $847, with higher costs driven by the addition of delamanid to replace pretomanid. Regimens using child-friendly formulations in the <5-year-old age group were marginally more costly than using adult formulations (median $142 range $122-$157), driven predominantly by cost differences for child friendly bedaquiline and delamanid compared to adult formulations as with other regimen categories. The reduced length of treatment of these regimens (6-months) offers potential for costs to drop below those for existing longer regimens, particularly if further price reductions in delamanid and bedaquiline are achieved.

### Individual medicine: Monthly costs

Representing per month medicine costs provides insights into the expected reductions that can be achieved by shortening regimens, and conversely by informing how extended regimens can substantially add to regimen costs.

**Table 4. Cost per month of RR-TB medicines (GDF prices and WHO dosing recommendations).**

| Medicine | Child-friendly formulations | | | Adult formulations | | |
|---|---|---|---|---|---|---|
| | Formulations | Monthly cost (US$) in children aged*: | | Formulations | Monthly cost (US$) in children aged*: | |
| | | <15 years | <5 years | | <15 years | <5 years |
| Bedaquiline: Month 1 *(existing dosing recommendations)* | 20mg tablet | 83.00 | 39.05 | 100mg tablet | 27.80 | 19.16 |
| Bedaquiline: Month 2 onwards *(existing dosing recommendations)* | 20mg tablet | 29.24 | 13.64 | 100mg tablet | 10.16 | 7.80 |
| Bedaquiline: Month 1 *(potential future once-daily dosing)* | 20mg tablet | 102.49 | 48.14 | 100mg tablet | 34.58 | 24.36 |
| Bedaquiline: Month 2 onwards *(potential future once-daily dosing)* | 20mg tablet | 68.24 | 31.82 | 100mg tablet | 23.71 | 18.20 |
| Levofloxacin | 100mg dispersible tablet | 14.76 | 6.24 | 250mg tablet | 1.45 | 0.68 |
| Moxifloxacin | 100mg dispersible tablet | 16.72 | 10.15 | 400mg tablet | 4.34 | 4.34 |
| Linezolid | 150mg dispersible tablet | 14.58 | 7.74 | 600mg tablet | 3.22 | 2.69 |
| Clofazimine | 50mg capsule | 13.74 | 8.12 | 100mg tablet or capsule | 12.12 | 7.00 |
| Cycloserine | 125mg capsule | 39.04 | 21.96 | 250mg capsule | 12.20 | 7.52 |
| Terizidone | 250mg capsule | 84.69 | 52.22 | 250mg capsule | 84.69 | 52.22 |
| Ethambutol | 100mg tablet | 5.20 | 2.63 | 400mg tablet | 1.41 | 0.60 |
| Delamanid | 25mg dispersible tablet | 155.69 | 99.01 | 50mg tablet | 102.37 | 95.75 |
| Pyrazinamide | 150mg dispersible tablet | 24.10 | 11.45 | 500mg tablet | 0.97 | 0.50 |
| Ethionamide | 125mg dispersible tablet | 12.05 | 6.78 | 250mg tablet | 4.43 | 2.73 |
| Prothionamide | 250mg tablet | 4.45 | 2.75 | 250mg tablet | 4.45 | 2.75 |
| Para-aminosalicylic acid (PAS) | 4g powder in sachet | 62.26 | 39.39 | 4g powder in sachet | 62.26 | 39.39 |
| Pretomanid~ | 200mg tablet | 36.56 | 36.56 | 200mg tablet | 36.56 | 36.56 |
| Isoniazid | 100mg tablet | 1.13 | 0.64 | 300mg tablet | 0.61 | 0.43 |
| Pyridoxine (Vit B6)~ | 50mg tablet | 0.22 | 0.18 | 50mg tablet | 0.22 | 0.18 |

*Costs reflect mean costs per regimen adjusted for distribution of weight-for-age in each category.

~ Same formulation used as child-friendly and adult formulation.

GDF-Global Drug Facility, US$-United States Dollar.

Table 4 details mean costs for one month's treatment of individual RR-TB medicines (using GDF prices and WHO dosing recommendations) for children aged <15 and <5 years old, and by child-friendly and adult medicine formulations. Mean monthly costs are weighted by the distribution of child weights across specific age ranges, and disaggregated medicine costs by weight band are presented in Appendix 2A and Appendix 2B in S1 File.

Monthly costs of delamanid were substantially higher than any other child TB medicine across formulation and age ranges, however bedaquiline, terizidone and para-aminosalicylic acid represent relatively high per monthly costs. Bedaquiline dosing during month 1 is higher than subsequent months due to the higher loading dosages. The adapted BPaL/BPaLM regimens use higher volumes of bedaquiline than currently recommended regimens, with subsequent impact on total regimen costs.

Fig 2 details the individual medicine cost contribution to different regimens. Delamanid represents an extensive cost component of both the shorter and longer regimens, while bedaquiline, cycloserine and clofazimine all represent significant cost drivers. Within adult formulations, medicines including levofloxacin, moxifloxacin, linezolid, ethambutol, pyrazinamide, ethionamide were all below $5 per month and make negligible contributions to the overall regimen cost.

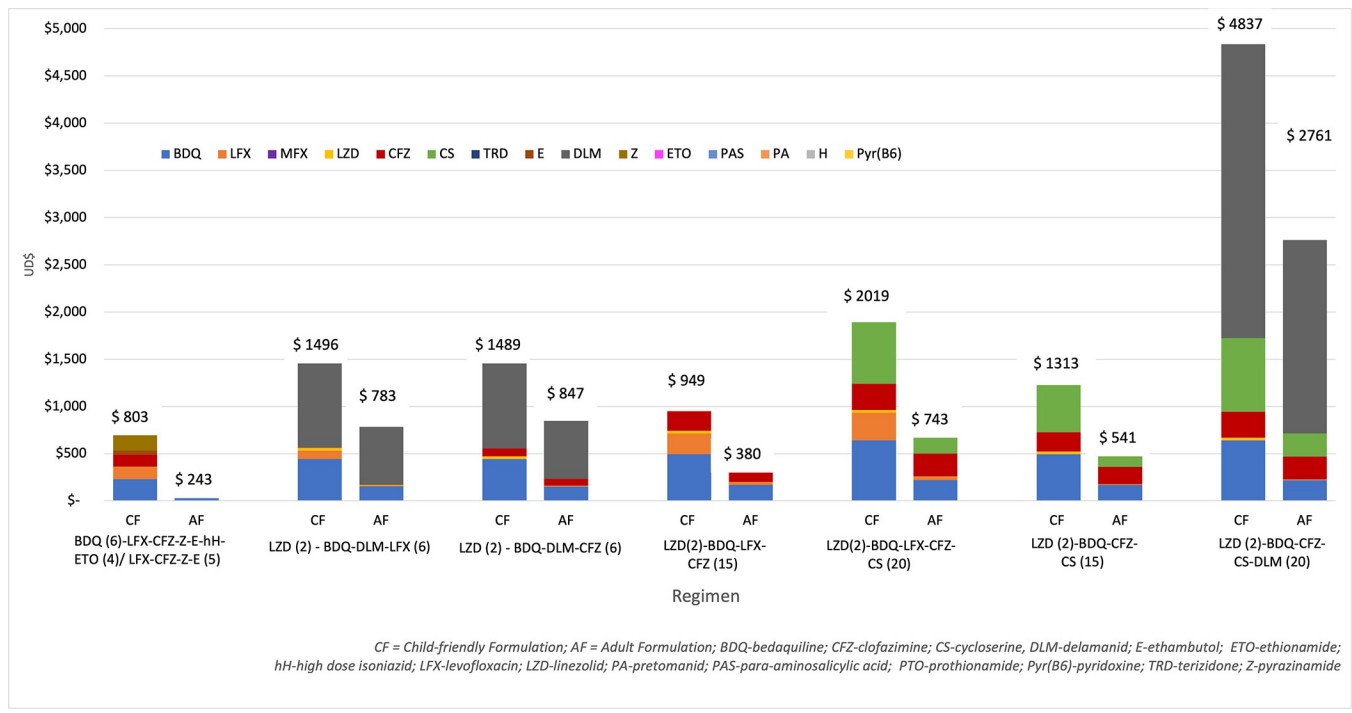

**Fig 2. Individual medicine cost contribution to total regimen cost (US$) in who-recommended RR-TB regimen in children and adolescents <15 years old (GDF pricing).**

Regimens using all-child friendly formulations were 1.5 times more expensive than all-adult formations in the <5-year-old age group, and 2.1 times more expensive in the <15-year-old age group. This is largely driven by price disparities related to the different formulations of bedaquiline; recent GDF procurement reduced the monthly cost for maintenance dosing of bedaquiline using adult formulations to $10, while the child friendly formulation to achieve the same dose costs approximately $30 per month [9]. The larger relative disparities–such as levofloxacin dispersible tablets incurring a monthly cost 9 times greater than the solid-dosage formulation—has less overall impact due to lower absolute monthly costs.

## Country comparison

A total of 24 regimens recommended under national guidance were analysed by regimen type (short, long), disease severity (non-severe, severe), fluoroquinolone-resistance profile and country (South Africa, India, the Philippines), see Appendix 4 in S1 File. National guidance generally tended to align with WHO, and tended towards longer regimen length which raised overall costs. Utilising South African national procurement prices, regimens were typically more expensive; the mean monthly cost of assessed adult formulations purchased through national procurement in South Africa were twice as high than if South Africa were to access the same formulations with GDF pricing. South African procurement does obtain some high-cost medicines at a lower price than the GDF; notably in the February 2024 contract listings, South Africa achieved a price for terizidone 250mg capsules at 44% lower cost than the GDF [20].

To demonstrate generalised local affordability of globally-recommended regimens in South Africa, India, and the Philippines, five regimens from WHO guidance were analysed adjusting for PPP and assuming GDF procurement arrangements for all settings (S1 Appendix 5 in S1 File). Regimens for children aged <15 years are presented in Fig 3 as a factor of the value of

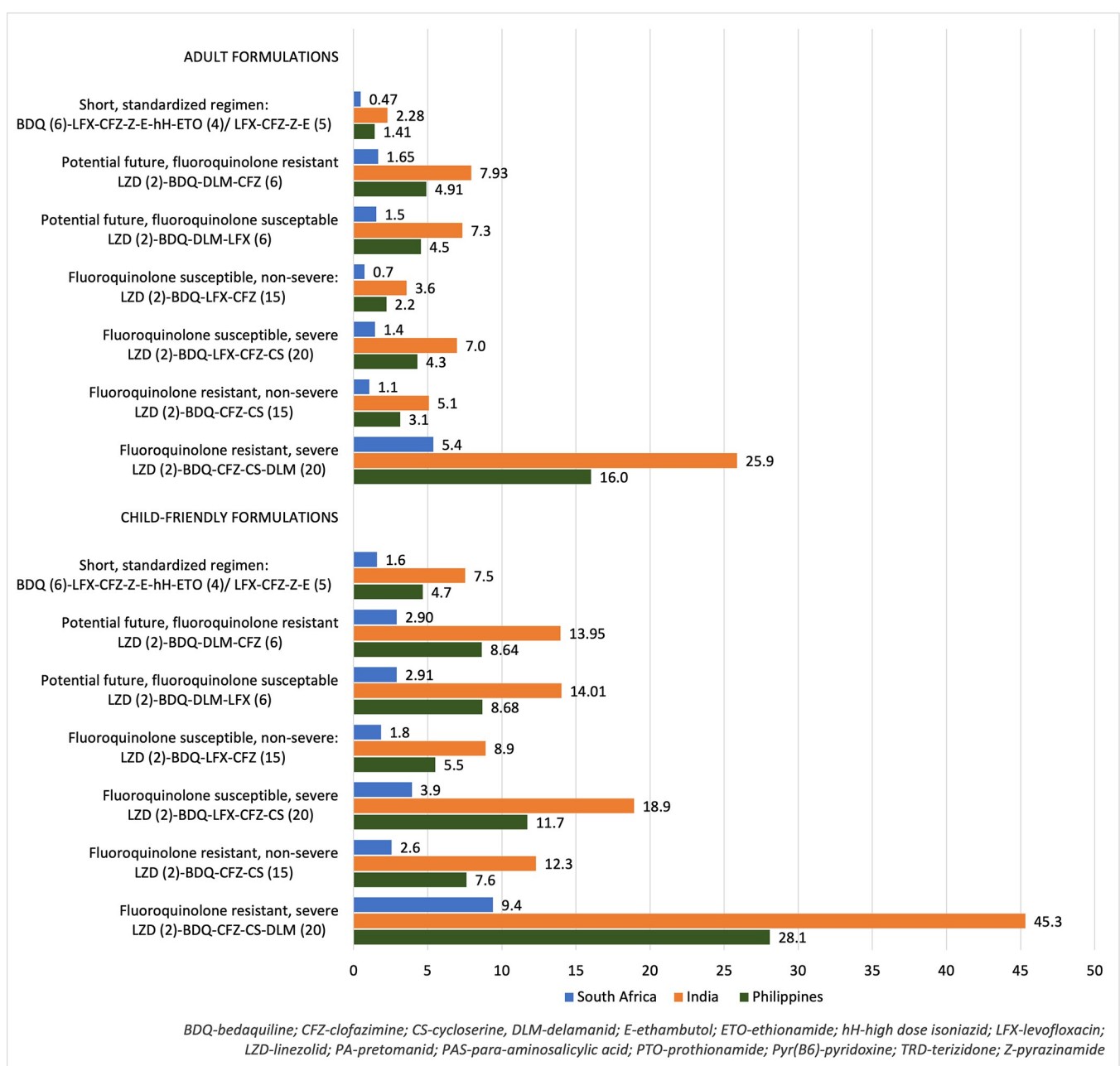

**Fig 3. Who RR-TB regimen cost as a proportion of national tb funding per tb case (GDF procurement, PPP adjusted).**

national TB program funding per incidence of TB, and in S1 Appendix 7 in S1 File as a factor of domestic government health expenditure for the three countries. The PPP-adjusted costs for each of the WHO RR-TB regimens (children aged <15 years) were higher than the total funding available per TB case in India and the Philippines, whereas the cost of the standardised 9-month bedaquiline-containing regimen cost approximately one third of funding available per case of TB in South Africa. The difference is most pronounced in higher cost regimens, where medicine costs for fluoroquinolone-resistant, severe disease in children <15 years old using adult formulations is 26 times more than the available funding per TB case in India, 16

times more in the Philippines, and five times more in South Africa. The mean PPP adjusted WHO RR-TB regimen costs for childen aged <15 years using adult formulations were 46, 15 and 3 times more than the domestic general government spending on health per capita in India, the Philippines and South Africa, respectively.

## Discussion

This research details the cost of various regimens for the treatment of RR-TB in children and adolescents <15 years age in the contexts of South Africa, India, and the Philippines. It is the first pharmaceutical costing analysis in childhood RR-TB that comprehensively incorporates individual medications over all weight distributions in the age group across all available child and adult formulations, and which details both international and national guidance.

The analytical findings are important for national TB programs in planning and budgeting for the management of childhood TB, and in cost effectiveness analyses, particularly relating to the adoption of new treatment regimens and new child-friendly formulations. The analysis supports existing research in this area, notably the Médecins Sans Frontières Access Campaign "DR-TB Drugs Under the Microscope" assessment of adult and child TB medicines more generally [30]. The analysis contributes to an extensive cost consolidation exercise conducted under the Global Health Costing Consortium, which reported a range of TB-related costs, but found limited child-specific costs and no reported costs for RR-TB in children [31].

The findings demonstrate the importance of the pricing of individual medicines and their impact on overall treatment costs. The unit price of delamanid, at a mean monthly cost of $102 and $156 in the <15-year-old age range using adult and child-friendly formulations respectively, drives the consistently high costs of all delamanid-containing regimens. The fluroquinolones (levofloxacin and moxifloxacin) are a mainstay of RR-TB treatment with emerging evidence for their use in prevention of disease following exposure to RR-TB in children and are some of the lowest cost RR-TB drugs at $1.45 and $4.34 per month respectively. Consequently, regimens treating patients with fluoroquinolone resistance are significantly more expensive as higher cost medicines are substituted into regimen.

The adapted BPaL/BPaLM regimens had comparable overall costs to currently recommend regimens; however, regimens with a combination of both delamanid and bedaquiline were relatively more expensive under current price structures. The 6-month treatment period of the adapted BPaL/BPaLM regimens is likely to result in lower overall health service costs with associated benefits for patients and carers. Although these regimens are not currently recommended in children, their introduction would have the potential to change the wider costing and economic landscape for RR-TB in children. Further cost analysis in this area will be a useful input to national and international recommendations about the adapted BPaL/BPaLM regimens in children.

The analysis highlights the relatively modest additional cost of adopting child-friendly medicine formulations compared to requiring children to use adult formulations. In particular, the difference in costs between adult and child-friendly formulations is lower in the <5-year-old age group, indicating that prioritising younger children for child-friendly formulations may be more affordable for national TB programs. The higher price for child-friendly formulations is expected to be associated with procurement volumes, and economies of scale associated with wider use of child-friendly formulations may lead to reduced prices through commercial negotiations and competitive contracting systems. Advocacy from the global community contributed to the 55% unit price reduction for bedaquiline in August 2023, however the price reduction was also likely enabled by the wider use of bedaquiline globally as the price reduction applied to the adult formulation of bedaquiline only [9]. Other factors that may contribute

to price reductions include cost-plus pricing estimations, where a price that reflects manufacturing, supply, and profit inform contract negotiations [32]. Under a cost-plus pricing approach, larger volumes and certainty of demand through pooled procurement is also expected to result in lower prices overall as production and supply costs are reduced at scale.

The approach to procurement is an important factor determining treatment costs, notably the finding that if South Africa were to adopt GDF pricing it could reduce expected pharmaceutical costs of RR-TB treatments in children substantially. The analysis demonstrates that it may be possible for South Africa to achieve a budget-neutral transition to all child-friendly formulation regimens by offsetting any increased expenditure with improved overall pricing of TB medicines through the GDF.

The substantial variation of costs in the analysis across contexts showed the limitations of representing common global costs of RR-TB treatment in children. The analysis showed that many factors are expected to influence costs, including differences in local guidance and prescriber choice, procurement practices and the prevalence of fluoroquinolone resistance. The concept of "severe" and "non-severe" RR-TB is an important driver of regimen choice and subsequently costs, and while in practice the severity of RR-TB disease cannot be neatly differentiated into two distinct groupings, this analysis demonstrates that models of care that can identify and treat patients before progression to more severe forms of disease are expected to have lower treatment costs, in addition to better outcomes expected from earlier treatment.

The country-specific analysis showed substantial variation in affordability of the different regimens in South Africa, India, and the Philippines, and demonstrated the acute budgetary need to achieve lower cost treatment options in fiscally constrained settings. Assessing expected budget impact relative to available funding is particularly relevant for national TB program decision-making but should also be a key consideration in the generation of global TB recommendations. As the incidence of RR-TB in children is relatively low, the total budget impact of RR-TB treatments in this group are expected to be only a small proportion of overall national TB program budgets. However, there is a continual need to ensure efficient spending of limited health and TB program funding. This analysis is highly applicable to national TB programs in South Africa, India, and the Philippines, and the approach is adaptable to other country TB programs, particularly those accessing GDF pricing.

This analysis assessed pharmaceutical treatment costs only. Additional health system and household costs, including diagnostics, hospital admission and outpatient visits, represent important factors relating to the affordability and feasibility of the management of childhood RR-TB. This is relevant when considering differences in total health system costs associated with child-friendly formulations, where there are potentially substantial cost savings and health benefits achievable though improved administration and adherence. A consistent finding in formulation acceptability studies is that child-friendly formulations are preferred by both children and caregivers [33]. Further, the routine use of adult formulations in children represents a risk of inconsistent and inaccurate dosing due to variation in approaches to administer medicines including halving, crushing, and dissolving tablets. It also often results in worse medication palatability and acceptability, a high priority patient-centered outcome, which may have negative impacts on adherence and child and caregiver experiences. Although the costs of these risks are difficult to quantify, a proportion of any higher costs associated with child-friendly formulations would be offset by expected improvements in dose accuracy. Lastly, the monitoring and management of adverse events associated with different regimens were not represented in this analysis and may have substantial cost implications that will vary by regimen used.

The global community is unlikely to reach the ambitious sustainable development goal to eliminate TB by 2035 without sustained action and innovation at critical points in the TB care

pathway. An essential enabler driving decision-making and action at international and local levels is a comprehensive understanding of costs and affordability of different medicines and combination regimens in key populations, particularly children.

## Supporting information

**S1 File. Pharmaceutical costs child MDR-TB.**
(DOCX)

## Author Contributions

**Conceptualization:** Thomas Wilkinson, Anthony J. Garcia-Prats, Mandar Paradkar, Aarti Kinikar, Melchor V. Frias, Edina Sinanovic, Anneke C. Hesseling, James. A. Seddon, Megan Palmer.

**Data curation:** Thomas Wilkinson.

**Formal analysis:** Thomas Wilkinson, James. A. Seddon.

**Funding acquisition:** Anthony J. Garcia-Prats, Tina Sachs, Anneke C. Hesseling, Megan Palmer.

**Investigation:** Thomas Wilkinson, Anthony J. Garcia-Prats, Edina Sinanovic, Megan Palmer.

**Methodology:** Thomas Wilkinson, Anthony J. Garcia-Prats, Tina Sachs, Anneke C. Hesseling, James. A. Seddon.

**Project administration:** Thomas Wilkinson, Anthony J. Garcia-Prats, Tina Sachs, Anneke C. Hesseling, Megan Palmer.

**Supervision:** Anthony J. Garcia-Prats, Mandar Paradkar, Aarti Kinikar, Melchor V. Frias, Edina Sinanovic, Anneke C. Hesseling, James. A. Seddon, Megan Palmer.

**Validation:** Anthony J. Garcia-Prats, Tina Sachs, Mandar Paradkar, Nishi Suryavanshi, Aarti Kinikar, Melchor V. Frias, Anneke C. Hesseling, James. A. Seddon, Megan Palmer.

**Visualization:** Thomas Wilkinson, James. A. Seddon.

**Writing – original draft:** Thomas Wilkinson, Anneke C. Hesseling.

**Writing – review & editing:** Thomas Wilkinson, Anthony J. Garcia-Prats, Mandar Paradkar, Nishi Suryavanshi, Aarti Kinikar, Melchor V. Frias, Edina Sinanovic, Anneke C. Hesseling, James. A. Seddon, Megan Palmer.

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
