## [Decision Letter · Decision Letter 0]

27 Mar 2024

PONE-D-24-06760Pharmaceutical cost dynamics for the treatment of rifampicin-resistant tuberculosis in children and adolescents in South Africa, India, and the PhilippinesPLOS ONE

Dear Dr. Wilkinson,

Thank you for submitting your manuscript to PLOS ONE. After careful consideration, we feel that it has merit but does not fully meet PLOS ONE’s publication criteria as it currently stands. Therefore, we invite you to submit a revised version of the manuscript that addresses the points raised during the review process.

We look forward to receiving your revised manuscript.

Kind regards,

Steve

Stephen Michael Graham, FRACP, PhD

Academic Editor

PLOS ONE

Journal Requirements:

Reviewers' comments:

Reviewer's Responses to Questions

**Comments to the Author**

1. Is the manuscript technically sound, and do the data support the conclusions?

Reviewer #1: Yes

Reviewer #2: Yes

2. Has the statistical analysis been performed appropriately and rigorously? 

Reviewer #1: N/A

Reviewer #2: Yes

3. Have the authors made all data underlying the findings in their manuscript fully available?

Reviewer #1: Yes

Reviewer #2: Yes

4. Is the manuscript presented in an intelligible fashion and written in standard English?

Reviewer #1: No

Reviewer #2: Yes

5. Review Comments to the Author

Reviewer #1: General comments:

1) Update references (and corresponding information and estimates), when appropriate (eg, Global TB report, latest GDF catalogue)

2) Carry out a detailed language check throughout (several typos, repetitions and other errors were found, some of which are mentioned below but not all).

3) In the introduction (or the discussion?), I would recommend adding a couple of sentences on the concept of cost-based generic prices and add a reference to Hill A, Barber M, Gotham D. Estimated costs of production and potential prices for the WHO Essential Medicines List. BMJ Global Health.

4) In the discussion, I recommend adding a couple of statements on the risk of using adult formulations (even if they may end up being cheaper for programmes) when they are manipulated for administration in children (under/over dosing etc).

5) Include considerations around “future regimens” (but I encourage you to change this terminology) in the discussion, as this part is totally missing now. Include some forward-looking considerations on the affordability of such regimens if recommended by WHO in the context of the other regimens you describe.

ABSTRACT

- “Providing child-friendly formulations of medicines in the <5-year-old age group across all WHO recommended regimen is expected to cost an additional $380 (range $212-$563) per child” – please, add some short text on the added value of investing in this despite the additional cost

INTRODUCTION

- Update reference 1 and estimated incident cases with data from the latest Global TB report 2023 (including for Table 1)

- Update reference 8 with the latest GDF catalogue available and update link too.

- Page 12, add a relevant reference to the CATALYST trial and very briefly provide an update on its status to better contextualize this specific study.

- Page 13, the sentence “Purchasing through GDF can result in reduced prices and improved continuity of supply than could be achieved through individual country procurement” – what is it meant with “continuity of supply than could be achieved through individual country procurement”?

- Page 13, second paragraph, rephrase with “The GDF catalogue includes formulations of TB medicines for adults and children, with pricing and formulation information regularly updated”

- Page 13, second paragraph – please provide updated estimates of countries procuring through national programmes, if possible. The current information you provide is from 2021, which is relatively old. Also, perhaps it would be good to add a couple of statements on the nuances between procuring first-line medicines (previously supported by donors and done via GDF by most countries, but now mostly procured with local budget and not anymore from GDF) and second-line medicines.

- Page 13, second paragraph – replace “only some are accessing child-friendly formulations” with “are procuring child-friendly formulations”

METHODS

- Page 14, first para – what do you mean with “ingredients-based pharmaceutical cost analysis”? Perhaps it becomes clearer later on in the paper, but at this stage it is not. It sounds as if you have done an estimate of how much a certain formulation should cost based on the cost of the API, manufacturing + some revenue for the developer (but I do not think this is the case).

- Page 14, first para – clarify what you mean with “from the perspective of national TB programmes in South Africa, India and the Philippines and at a global perspective”?

- Page 14, second para – propose rephrasing as “The RR-TB treatment regimens utilized in this cost-analysis included regimens as per global (…) and national (…) guidance, and potential future regimens based on authors’ judgement”

- Please, check the spelling of the methods section in general, eg, “severe regimens” does not make too much sense; “The BPaLM/BPaL (please, invert the order) adaption regimens (please, rephrase).

- Page 15 – what does it mean that “The BPaL/BPaLM adaption regimens are not currently recommended by WHO or any national TB treatment guidance for children “but reflect the opinions of a wide group of experienced child TB physicians” (please, rephrase, ie “are provided for comparison as potential future treatment scenarios, in line with experienced child TB physicians’ opinion on potential treatment regimens available for children in the future..)

DATA ANALYSIS

- Page 16 – “Therefore, GDF pricing was used as indicative in the Indian context, under

an assumption that prices are expected to be comparable between the Central TB Division and the GDF procurement mechanism” – what information you have to assume that prices are expected to be comparable between the Central TB Division and GDF? Is this assumption based on specific criteria, for example the high medicine volume procured by both india and GDF which could lead to the same prices? Please, clarify.

- Page 18 – “WHO formulation dosing recommendations were used to determine assumptions…” Please, rephrase.

RESULTS

- Page 19 – what do you mean with “given limited specifications for these formulations in national guidance in South Africa, India, and the Philippines”? Does this relate to the lack of clarity on whether these paediatric formulations are available in the respective countries? Country guidelines usually do not include information on whether paediatric formulations of TB medicines are used or not. I strongly recommend looking into the national Essential Medicines Lists for these countries or retrieve this information elsewhere.

“Standardised short all-oral regimen”

- Page 21 – “The standardised “short” regimens (9-11 months) are currently the preferred choice of treatment for RR-TB according to WHO as well as South African and Indian national TB guidelines for patients with uncomplicated, non-severe, fluoroquinolone-susceptible RR-TB disease”. According to WHO and if you do not specify the age, the preferred regimen is BPaLM/BPaL. Please, clarify this sentence possibly by specifying the age you refer to.

-

DISCUSSION

- Page 29, second paragraph “particularly relating to the adoption of new treatment

regimenS and new child friendly formulations.”

- Page 29 – repetition of “this analysis” multiple times across these paragraphs – check language and rephrase.

- Page 29, second paragraph “The analysis supports complements existing analysis in this area, notably the routine pricing analysis under the Médecins Sans Frontières Access Campaign “DR-TB Drugs Under the Microscope” that assesses the cost of adult and child TB medicines and regimens more generally (29)”

- Page 30, first paragraph “replace national treatment programs” with “national tuberculosis programmes”

- Page 30, “The child-friendly formulation price differential is also a direct consequence of procurement practices” – does this refer to what you specify in the following sentence, that is the it depends on the volumes procured? If this is the case, then “procurement practices” is not the best terminology to use. Please, rephrase with, eg, procurement volumes.

- Page 30 “it can be expected that economies of scale will lead to reduced prices, demonstrated by a 55% price reduction in the price of bedaquiline in August 2023 that

applied only to the adult formulation (8)”. I would smoothen the language behind this statement. In general, it is true that as volumes increase, then you can negotiate better prices, but that’s not always the case. Plus, for bedaquiline, there were also strong advocacy pushes to lower the price (it is not just the increased volumes).

- Page 32 – add a reference to back up this statement: “A consistent finding in formulation acceptability studies is that child friendly formulations are

preferred by both children and caregivers”.

Reviewer #2: Dear Authors,

While a straightforward concept, this article presents useful information on drug costing in children that may be useful to national TB programs to inform planning and budgeting.

As mentioned in the discussion, the article could have been strengthened by a more comprehensive analysis on total costs e.g. costs to conduct basic minimum active drug safety monitoring (laboratory testing, ECG etc) and hospital costs (outpatient/hospitalisation etc).

The article is otherwise clearly written and presented.

As a minor point, references could also be updated e.g. 2023 global TB report instead of 2022 (ref 1). Also page 14: it is not clear to me that costs were "modelled" ? were they not just calculated based on regimen and formulation costs.

6. PLOS authors have the option to publish the peer review history of their article (what does this mean?). If published, this will include your full peer review and any attached files.

Reviewer #1: No

Reviewer #2: No

---

## [Author Response · Author response to Decision Letter 0]

24 May 2024

Thank you for the reviewer comments. We consider that we have addressed all comments sufficiently as detailed in our response letter uploaded. However we would welcome the opportunity to make any further changes as required.

---

## [Editor Report · Decision Letter 1]

7 Jun 2024

Pharmaceutical cost dynamics for the treatment of rifampicin-resistant tuberculosis in children and adolescents in South Africa, India, and the Philippines

PONE-D-24-06760R1

Dear Dr. Wilkinson,

Thanks for resubmission with improvements to the manuscript. We’re pleased to inform you that your manuscript has been judged scientifically suitable for publication and will be formally accepted for publication once it meets all outstanding technical requirements.

Kind regards,

Steve

Stephen Michael Graham, FRACP, PhD

Academic Editor

PLOS ONE
---

## [Editor Report · Acceptance letter]

15 Jul 2024

PONE-D-24-06760R1 

PLOS ONE

Dear Dr. Wilkinson, 

I'm pleased to inform you that your manuscript has been deemed suitable for publication in PLOS ONE. Congratulations! Your manuscript is now being handed over to our production team.

Kind regards, 

on behalf of

Dr. Stephen Michael Graham 

Academic Editor

PLOS ONE